# Privacy and Auditability in the Local Energy Market of an Energy Community with Homomorphic Encryption

Davide Strepparava [1] , Federico Rosato [1,2,*] , Lorenzo Nespoli [1] and Vasco Medici [1]

1 Department of Environment Construction and Design, University of Applied Sciences and Arts of Southern Switzerland (SUPSI), 6850 Mendrisio, Switzerland; davide.strepparava@supsi.ch (D.S.); lorenzo.nespoli@supsi.ch (L.N.); vasco.medici@supsi.ch (V.M.)
2 Department of Energy, Politecnico di Milano, 20156 Milan, Italy
* Correspondence: federico.rosato@supsi.ch

**Abstract:** The world of electrical distribution is rapidly changing and is seeing more and more distributed production and steerable flexibilities. Energy communities are seen as an important innovation for the optimization of electrical consumption at a local level. A central need of the local energy markets inside energy communities is the exchange and circulation of production and consumption data, and therefore the problem of the potential leak of sensitive data must be addressed. In this paper, the context of the Lugaggia Innovation Community, a Self Consumption Community pilot project in southern Switzerland, is introduced together with the blockchain framework that was created for its internal market interaction and the rules designed for its local energy market. A cryptographic protocol from the literature, based on homomorphic encryption, is then proposed for the anonymous aggregation of production and consumption data of the individual households at a resolution of 15 min. The computational overhead associated with the protocol is then experimented and analyzed.

**Keywords:** energy community; smart grid; local energy markets; homomorphic encryption





## 1. Introduction

The importance of energy communities (ECs) for tackling the challenges that the electrical distribution grids will face in the future is being more fully recognized. Pilot projects are particularly important, as they allow for testing and implementation of solutions in a real setting, highlighting successful methodologies and potential problems. The Lugaggia Innovation Community (LIC) was born as a Self Consumption Community (SCC), with the sponsorship of the local distribution system operator (DSO), Azienda Elettrica di Massagno (AEM). For this energy community, a local energy market (LEM) was established, backed by a blockchain framework based on Cosmos [1], as extensively introduced in [2]. Although the data exchanged in the energy market are intrinsically pseudonymous, the danger of leaking sensitive data persists, and it must be solved to allow for the replicability of the community model. A simple encryption solution for the sensitive data, however, would undermine the auditability of the market. Therefore, a protocol for privacy preservation based on homomorphic encryption was selected, implemented, and tested on the hardware of LIC. The protocol allows for the market settling to be auditable by any interested party without publishing any plain text data.

## 2. European and Swiss Regulatory Context

In this section, the regulatory contexts in which ECs exist are briefly reviewed, mainly with reference to the European Union (EU) and Switzerland. ECs, as organizations, are subject to general data protection requirements such as the General Data Protection Regulation (GDPR) for the EU or the Federal Act on Data Protection (FADP) for Switzerland. We then make a few considerations on additional steps to be taken to ensure compliance.

## 2.1. Energy Communities

The term "energy community" is routinely used to indicate a vast set of realities concerning groups of subjects that agree on some form of organization to satisfy some of their energy-related needs [3]. As a generic concept, associations for sharing and use of energy resources have existed for a long time, in particular, in remote inhabited places where they represented the only viable solution for energy supply [4]. In recent times, the energy community was recognized as an important and strategic piece of the energy transition, with a strong accent on renewable energy sources (RES) [5]. The EC is a framework that promotes local and rational use of energy resources. The chief concrete advantage of the EC is the facilitation of self-consumption, which is positive both from an economical perspective for the participants and from a systemic one, as it maximizes the decarbonization potential of the distributed RES [6] and the opportunity to use smart control to relieve the electrical distribution grid [7]. Among the energy assets to be shared, of particular importance are the ever-increasing amount of Photovoltaics (PV) panels installed at the residential level and, incipiently, electrical vehicles (EVs), that represent an abundant but time-flexible electrical demand and even "strategic" storage with novel vehicle-to-grid (V2G) capabilities [8]. A key feature that makes the energy community an important transition instrument is, moreover, its social impact value. As observed in [9], *"behaviour change will likely occur in the context of changing values, lifestyles, and cultural norms modulated through social contexts"*. The EC provides a comprehensive framework in which, as observed in [10,11], the direct involvement of the citizens in the decision process, and engagement in the operation of the energy assets, can foster acceptance and overcome resistance to the implementation.

Regulatory entities are trying to capture the potential of ECs. In 2016, the European Union presented the *Clean Energy for all Europeans Package*, commonly known as *Clean Energy Package* (CEP), a set of eight legislative acts constituting a substantial update of the European energy policy, aimed at easing the energy transition. Of particular importance for the EC theme are the 2018/2001 directive (RED II) and the 2019/944 directive (IEM), introducing, respectively, the renewable energy community (REC) and the citizen energy community (CEC). The REC is of particular comparative interest for the present work, as it is nearest to how the LIC community is organized. In the RED II directive defining the REC, the EU placed particular stress on the no-profit nature of these legal subjects, which should be chiefly aimed at providing social, economic, and environmental benefits. The participation is open and voluntary, they can manage several forms of energy (electricity, heat, gas), and they have a geographical proximity requirement for the generation plants.

Swiss law has mentioned self-consumption since April 2014, with the revision of the energy law then in place (Energiegesetz) [12]. A major update came with the new 2018 Federal Energy Law [13], which defines a new form of energy community: the ZEV (Zusammenschluss zum Eigenverbrauch), or RCP (Raggruppamento ai fini del Consumo Proprio) in the Italian language. With respect to the European legal framework, this entity is explicitly centered on the sharing of solar PV energy and can be constituted if the production capacity exceeds 10% of the total nominal connected power. Furthermore, the main stated motivation for the introduction of the ZEV is precisely the ability to seamlessly consume locally produced solar PV energy and sell only the excess for economical reasons (directly save on the purchase of electricity instead of losing the price difference with a sell–buy). It is explicitly stated that there must be one physical coupling point between the ZEV and the rest of the grid, and metering is done by the DSO only at this interface, such that the ZEV is considered a unique customer by the DSO. This means that:

- Metering and billing is the responsibility of the ZEV, which is free to choose its own hardware and billing procedures, subject only to generic law requirements;
- The ZEV is in charge of the distribution of electricity inside the ZEV itself, and therefore ownership and operation are separated from the rest of the distribution grid.

In contrast, for example, in the Italian reality, the EU directives were integrated with the D.Lgs. 162/19 and "virtual" self consumption using the existing grid owned and

operated by the DSO is allowed, as is also mentioned in the 318/2020 directives by the local competent regulation authority ARERA. This allows one to save on potentially avoidable modifications on the grid structure.

*2.2. Open Participation and Data Protection*

In light of the above review, it emerges how open participation is a very important aspect for ECs, as it is the driver of citizen engagement, necessary for fostering the energy transition. Open participation, however, is not without its dangers, in particular when there is an exchange of information. In a regulatory context like the Swiss one, where the EC is granted ample freedom in how the economical transactions are carried out internally, particular attention has to be paid both to the privacy of the users and the easy auditability of the correct execution of the agreed-upon mechanisms. In particular, ECs manipulate personal data and therefore are subject to data protection regulations.

The General Data Protection Regulation (GDPR) is a European Union law that was implemented on 25 May 2018, and it requires organizations to safeguard personal data and uphold the privacy rights of anyone in EU territory (https://gdpr.eu/faq/, accessed on 25 July 2022). The GDPR has been defined as the "toughest online privacy rules in the world" [14]. As observed in [15], *"The GDPR expands the scope of data protection so that it applies to anyone or any organisation that collects and processes information related to EU citizens, no matter where they are based or where the data is stored"* and, therefore, ECs fall under its scope, in that they intrinsically manipulate energy consumption data. Similarly, the 2020 revision of the Swiss Federal Act on Data Protection (FADP) aimed to enhance data protection in Switzerland, with explicit reference to the intention to be uniform with the surrounding countries (in order to ensure that European legislators and commercial partners will continue to consider Switzerland as a third party with adequate data protection policies [16]): *"The FADP provides an overall framework and deals with data protection using principles similar to those applied in other countries [...] The FADP is very wide in its scope and applies to personal data file activities carried out by Federal authorities, private organisations and individual private persons"* (https://www.edoeb.admin.ch/edoeb/en/home/the-fdpic/legal-framework/ii--a-few-facts-about-the-federal-act-on-data-protection.html, accessed on 25 July 2022).

In communities such as LIC, which, as explained below, is based on an open blockchain platform, the problem of data protection is particularly pressing. The transactions related to market execution involve individual energy consumption data and are, in principle, visible by anyone. Therefore, a suitable encryption scheme must be put in place to grant GDPR/FADP compliance. An immediate measure that could be taken is to use an encryption scheme and exchange only the ciphertexts associated with these data, with only the administrator holding the decryption keys (stored and handled in a GDPR/FADP compliant way) in order to settle the market. In this way, however, the auditability of the execution of the market mechanism (that is explored extensively later on) is no longer granted. The solution presented in this paper offers maximum user privacy while retaining the ability to audit the correctness of market execution.

## 3. The LIC Energy Community

The LIC pilot project consists of an EC created in Lugaggia, a small town near Lugano in the southern part of Switzerland. The community comprises 18 residential houses, of which four are equipped with roof photovoltaic panels for a total of 37 kWp, and a kindergarten, in which a battery with a capacity of 60 kWh and a 27 kWp photovoltaic system were installed. The community was created with the collaboration of the local DSO: AEM. The houses participating in the pilot project and the kindergarten were wired together under a single substation; the ownership of the underlying electrical network was split off from the DSO. Therefore, the LIC constitutes a proper, physically connected energy community. The collocation and map of the community are shown in Figure 1.

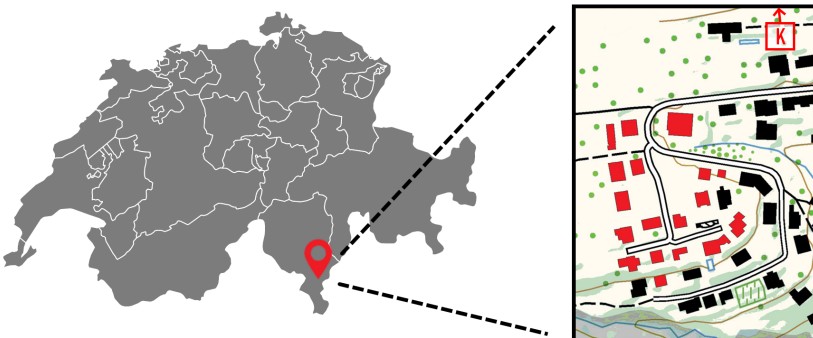

**Figure 1.** Geographic collocation and map of the LIC energy community in Lugaggia, Ticino, Switzerland. The buildings participating in the community are highlighted in red. The kindergarten is just above the view, indicated with a "K".

The underlying legal entity for the LIC pilot is that of an RCP under Swiss regulation. This association form has been possible in Switzerland since 2018 with the reform of the "Energy Law". Under the constraint of geographic contiguity, producers and consumers can come together to self-consume their own, locally produced renewable energy. The main research-wise aim of the project is validating algorithms and innovative methods for steering loads and flexibilities, in order to maximize the usage of the renewable energy. Aside from the battery installed on the kindergarten, the main steerable assets are heat pumps and electrical water heaters, whose load can be shifted without necessarily impacting user comfort by taking into account the appropriate availability constraints.

### 3.1. Motivation

The municipality of Capriasca installed the aforementioned 27 kWp PV plant in the village of Lugaggia on the roof of the local kindergarten. The building is located on the edge of a residential area, mainly consisting of single-family houses. The self-consumption potential of the kindergarten is limited because most of the production takes place during school summer holidays when the local consumption is low. AEM, the DSO serving the area, promoted the creation of a SCC (subsequently ZEV), connecting together the kindergarten and 18 nearby houses. The energy exchange inside the community is compliant with existing laws regulating the SCCs.

The district-level storage system was installed by AEM to further increase the flexibility of the SCC. The project aims to:

1. Evaluate the needs and requirements to the realization of LIC in a real environment. Provide recommendations on how to allow and facilitate the replicability and scalability of peer-to-peer SCCs.
2. Assess blockchain as a decentralized billing management method introduced by the utility.
3. Compare centralized vs. decentralized load management methods from the DSO point of view (grid costs), from both an energy consumption and economic standpoint.
4. Help to assess the local flexibility potential and the different technical means in which it could be exploited.
5. Evaluate the degree of knowledge, acceptance, and willingness to participate in a SCC among the community stakeholders.

### 3.2. Hardware

From a hardware standpoint, each of the delivery points is equipped with a smart meter. A programmable computing platform is connected to the smart meter through an IEC 62056-21 compliant optical USB meter reader with a rated communication speed of 19200 baud (REDZ KMK-116). The computing unit chosen was the Strato Pi CM, a compact industrial platform based on the Raspberry Pi Compute Module. The Strato Pi CM can be thought of as an industrial-grade version of the more common Raspberry, uniting the

ease of use and rich development possibilities offered by a complete Linux operating system running on an ARM v8 architecture with the reliability and high availability of an industry-grade electronic platform. Reliability-wise, a key feature of the computing unit is to offer a robust internal eMMC flash memory (instead of relying on an SD card) and a hardware watchdog. Internet connectivity for the computing units is granted by a USB dongle offering a mobile 4G connection. An installation is shown in Figure 2.

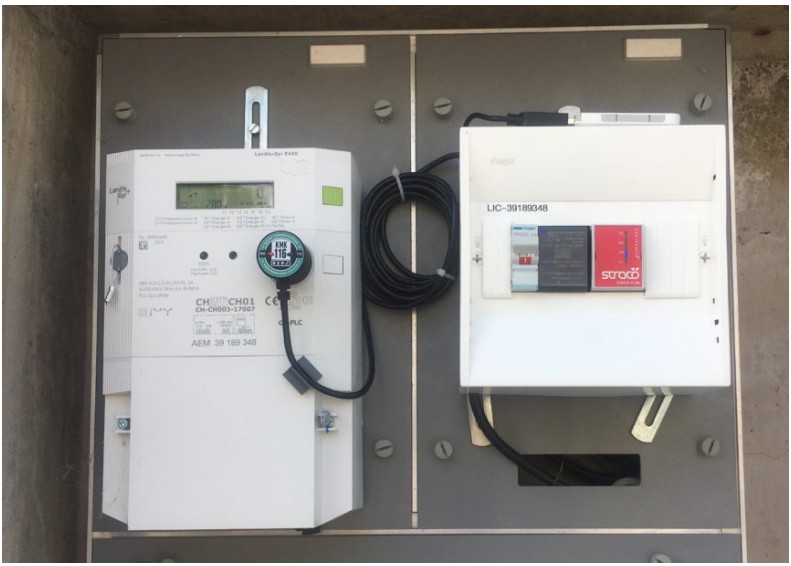

**Figure 2.** Strato Pi CM (**right**), USB 4G connection dongle (on the case of the Strato Pi CM) and smart meter (**left**) with optical reading probe, installed at a point of delivery.

## 4. Blockchain-Based Market Implementation

In order to provide a decentralized and trustless framework for the execution of the market and for the economic transactions, blockchain technology was selected. Albeit offering the ideal setting for a P2P energy market [17], solutions based on a plain blockchain implementation suffer from scalability issues and high hardware resource requirements, particularly for memory usage. To tackle these challenges and make distributed applications available to a broad range of IoT applications with high transaction throughput, in recent years, *second-layer approaches* were developed. One of the possible means of realization of a second layer is the *sidechain*, which is essentially a secondary blockchain whose consensus mechanism is linked to a parent chain. Sidechains internally validate their own transactions, turning validation over to the parent chain only in case of conflicts, and therefore remove a significant part of the strain associated with the sidechain application off the parent chain, allowing scalability. Sidechains are typically based on consensus algorithms that do not require significant computational power, which is an advantage compared to proof-of-work based blockchains from the point of view of sustainability. Data (e.g., tokens) can be transferred between a parent chain and a sidechain. A sidechain can be used to build custom solutions, taking into account the internal exchange of any data, not necessarily limited to the ownership and exchange of tokens. Thus, this technology is promising in LEM management.

After an exhaustive analysis of the state-of-the-art, the Cosmos project [1] was selected. Cosmos provides a complete environment to create custom sidechains. The Tendermint (https://tendermint.com/, accessed on 25 July 2022), a byzantine-fault-tolerant (BFT) [18] algorithm based on the proof of stake (PoS) mechanism, controls the consensus among the nodes of a zone. Nodes in Tendermint have non-negative voting power, and nodes with positive voting power are known as validators. Validators contribute to the consensus process by disseminating cryptographic signatures, in order to reach an agreement on the next block. The protocol needs to use a known set of validators, each of which is recognized by its public key. Validators try to reach an agreement on one block at a time.

A block is a collection of transactions. Rounds of voting are used to reach a consensus on a block. Each round has a round leader, also known as the *proposer*, who proposes a block. The validators then vote on whether to approve or reject the proposed block. The proposer of each round is chosen deterministically from an ordered list of validators according to their voting strength.

*Development and Deployment*

The second-layer solution based on the implementation of a sidechain has been developed, deployed, and tested in the LIC pilot during 2020 and 2021. The first activities relevant for the present work conducted in the LIC project can be divided into two parts: the deployment of the sidechain on LIC devices and the development of an application that uses the deployed sidechain and preserve the users privacy with pseudonimity. The following paragraphs describe them.

On the Strato devices, a Cosmos sidechain was deployed and is continuously running. Figure 3 shows the structure of the network. Each node corresponds to a device (there are 20 nodes because there is an additional, more powerful computing unit in the kindergarten used for other analyses, which partakes in the sidechain but does not correspond to an additional POD). In order to enhance communication security, the communication among the nodes is enclosed in a VPN.

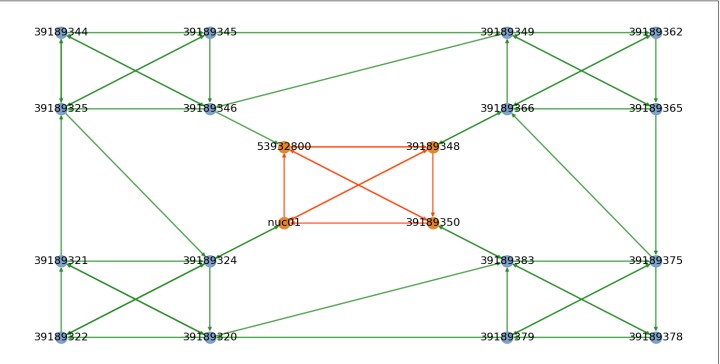

**Figure 3.** LIC network structure.

In Figure 3, the light blue circles represent nodes without specific permissions; the orange ones are the chain validators, i.e., the unique elements in the network allowed to create new blocks. These nodes have a relevant role inside the network. Indeed, if all the validators are offline (e.g., for a power outage or a connectivity problem), then no block will be created. To minimize the probability of such an event occurring, a collection of four nodes have been configured as validators.

## 5. Electricity Market

The goal of the community is to maximize its welfare by reducing the costs for the consumers and increasing the revenues of producers. At the same time, we want to formulate a market that meets the following desirable conditions:

- A fair redistribution of money among the market players, according to their contribution to the market's intended outcome;
- Variance reduction in the aggregated power profile;
- Compatibility with the current legal energy billing framework, considering both produced and consumed energy in a given time slot;
- Promotion of self-consumption at the community level, while retaining the ability to steer the overall power profile at the will of a third party (administrator).

We set up an automated market making (AMM) mechanism [19,20] that achieves the objectives above, defined by a set of simple and easily interpretable price formation rules:

- The energy consumed from the external grid shall be paid for as if the consumer were not part of the community;
- The energy consumed from inside the community is paid for at a total price lower than the standard tariff of the energy supplier and DSO, with a discount proportional to the ratio of the total produced and consumed energy;
- The energy injected into the external grid shall be remunerated as if the consumer were not part of the community;
- The energy injected that is consumed inside the community is remunerated at a price higher than the standard tariff of the energy supplier, with a discount proportional to the ratio of the total consumed and produced energy;
- The self-consumed energy is equally split among the community members proportionally to their consumption and production;
- The instantaneous buying and selling prices are dynamic, but for a given time slot, they are the same for everyone;
- The difference between the community buying and selling prices covers the cost to set up, operate, and maintain the community infrastructure.

These AMM rules can be mathematically expressed in the following way:

$$p_b = \left[ E_c p_b^{BaU} - \min(E_c, E_p) \left( p_b^{BaU} - p_b^{P2P} \right) \right] / E_c \tag{1}$$

$$p_s = \left[ E_p p_s^{BaU} - \min(E_c, E_p) \left( p_s^{P2P} - p_s^{BaU} \right) \right] / E_p \tag{2}$$

where $p_b$ and $p_s$ are the buying and selling prices generated by the AMM, $E_c$ and $E_p$ are the sum of the energy consumed and produced inside the energy community, and $p_b^{BaU}$, $p_s^{BaU}$, $p_b^{P2P}$, and $p_s^{P2P}$ are the buying and selling prices in the business as usual (BaU) case and those established inside the energy community (P2P). Peers clearly profit from the difference in price between BaU and P2P, but the third party also earns money when energy is self-consumed inside the community. It is important to notice that the P2P tariff is applied only to the energy produced by the members of the community, and as a consequence it is also in the third party interest to maximize self-consumption (no conflicting interests between peers and community admin). The AMM mechanism dictates the price formation inside the community. The prices as a function of the consumed and produced energy inside the EC can be seen in Figure 4. It can be shown that these prices generate convex costs as a function of the agents' actions, and thus they are amenable to be jointly optimized in a distributed way, as in [21,22].

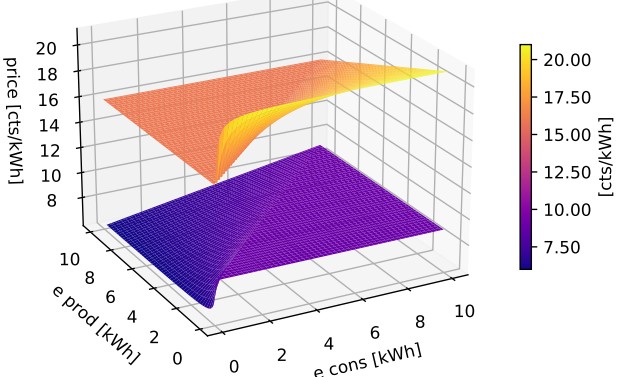

**Figure 4.** Buying (**upper surface**) and selling price (**lower surface**) $p_b$ and $p_s$ as a function of the produced and consumed energy inside the EC.

Let us consider the cost of the $i$th agent when consuming energy inside the EC. If the agent changes its consumption $e_i$, it directly influences $p_b$, as its consumption is included in $E_c$. Due to the presence of the *min* operator, we must split the $p_b$ expression into two

cases to study its convexity, according to whether $E_p$ is bigger or smaller than $E_c$. In the first case, the EC is a net energy producer, and the $i$th agent's total costs can be expressed (simplifying the above expression for $p_b$) as:

$$c_i = p_b e_i = p_b^{P2P} e_i, \tag{3}$$

which is linear; in the second case, the EC is a net energy consumer, and the expression of $p_b$ reduces to:

$$c_i = p_b e_i = p_b^{BaU} e_i - E_{p,0} \frac{p_b^{BaU} - p_b^{P2P}}{E_{c,0} + e_i}, \tag{4}$$

which is convex in $e_i$. Here, $E_{c,0}$ is the initial consumed energy of the EC, before the influence of the $i$th agent; similarly, $E_{p,0}$ is the produced energy before the production of the $i$th agent (in this case fixed at 0). Because the $i$th agent can switch the EC from being a net energy importer to being a net energy exporter, the two expression must be combined to study the overall convexity of the $i$th agent costs.

Figure 5 shows the combination of the two expressions. We fixed $E_{c,0}$ and $E_{p,0}$ to 1 and 5 kWh, respectively, and spanned the consumption $e_i$ of the $i$th agent from 0 to 10 kWh, so that the EC transitions from net energy producer to net energy consumer in the graph. The blue line shows the linear price that is generated in the case when the EC is a net energy producer, the orange one is the price when the EC is a net energy consumer, and the dashed green line is the true cost. As the true cost is the maximum of two convex expressions, it is also convex due to the convexity rule of composite convex functions. Similar reasoning can be done for the case in which the agent is a producer, and the same conclusion can be reached. Thus, the costs function is convex with respect to the agents' actions.

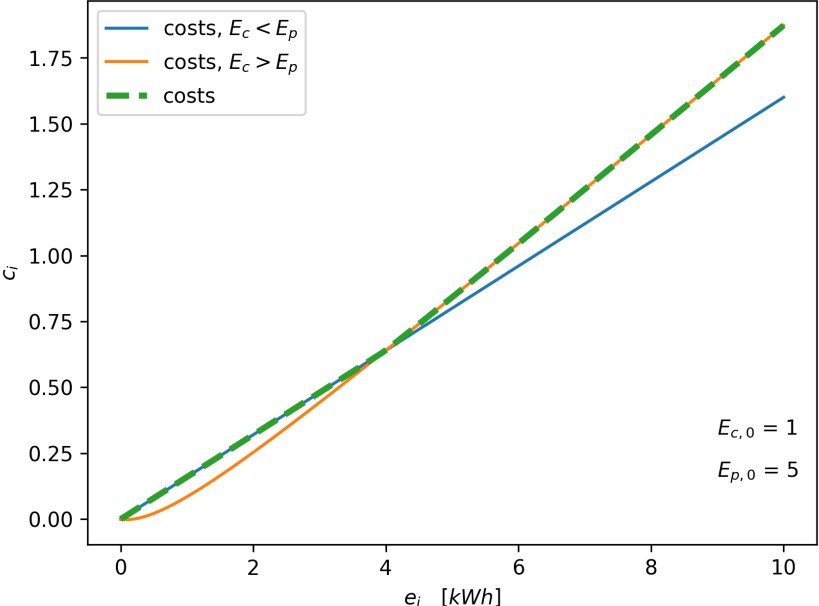

**Figure 5.** Buying cost (in green) as a composition of the two sub-cases in which the EC is a net energy importer (orange) or net energy exporter (blue). The shape of the lines depend on the fixed values of $E_{c,0}$ and $E_{p,0}$, but the final expression is always convex.

## 6. Full Anonymization of Participants

Data about the energy exchange of households, and in particular energy consumption data, carry sensitive information about consumers [23]. Conclusions regarding employment status, presence of specific appliances, daily routine, and economic status can be drawn from consumption data. In Equations (1) and (2), the quantities $E_c$ and $E_p$ are derived as the aggregation of the individual energy consumption measure posted by the certified meters.

These measures could be saved in plain text on the blockchain under a pseudonym (the blockchain address associated to the meter). Even so, however, the leak of sensitive data is still significant, and by analyzing the action on the market it would be conceivable to associate the pseudonym to the real user, for example, by matching production data with a PV installations database, or by minimally intrusive observation of presence/absence patterns. The individual bids could also be encrypted with common symmetric encryption, with the admin holding all the keys for decrypting the bids and performing the solution of the market and the billing. In this way, however, the individual users do not have the ability to verify the correctness of $E_c$ and $E_p$, having to trust the community admin on the correct price calculations. In order to be able to encrypt the on-chain bids but still retain the ability to audit the clearing of the market, in this work we used a scheme based on homomorphic encryption.

### 6.1. Homomorphic Encryption Protocol

Stated generally, homomorphic encryption is a form of encryption where some form of computation, for example, the sum, can be performed directly on the cyphertext, obtaining a new cyphertext that can be decrypted to obtain the final result of the applied operation [24]. The fact that the cyphertexts retain the ability to be manipulated with some operation without breaking the security of the data unlocks several possibilities for secure aggregated computations, even externally sourced. The idea of adding a homomorphic encryption layer to aggregation-based blockchain applications for enhancing privacy is explored, for example, in [25]. In [26], the authors exploit this fundamental property to propose a small scale, auditable "boardroom voting" (or, more in general, aggregation) protocol that ensures maximum user privacy. We adopted this scheme for the implementation of anonymization in LIC. The scheme works as follows. It is assumed that a secure broadcasting communication channel is available to the $n$ participants, in our case constituted by the Cosmos sidechain. Before any voting round takes place, the participants agree on a cyclic group $G$ with generator $g$ of prime order $q$ such that the decisional Diffie–Hellman (DDH) assumption [27] is considered to hold. Then, to realize a voting round:

1. The admin updates the list of the participants to the aggregation round and broadcasts it. With this information, each agent will be able to know when it has received all the information broadcast by the other participants at each step, so that it knows when to proceed with the following step.
2. Each participant decides on a "vote" to express, that is, a private integer value $v_i$.
3. Each participant selects a random integer value $x_i \in 0...q$, and therefore a $g^{x^i} \in G$. The value $\gamma_i = g^{x^i}$, termed *registration key*, is broadcast.
4. Each participant can reconstruct a list of keys, one for each participant $i$, with the following formula: $Y_i = \prod_{j=1}^{i-1} \gamma_i / \prod_{j=i+1}^{n} \gamma_i$. Notice that the calculation of $Y_i$ requires the knowledge of the registration keys of all the other participants. Being calculated by repeated multiplication or inversion of elements of $G$, $Y_i$ is still an element of $G$, and is therefore representable as $g^{y_i}$. The above formula ensures structurally that $\sum_i x_i y_i = 0$.
5. Each voter broadcasts their hashed vote $\Gamma_i = g^{x_i y_i} g^{v_i}$.
6. The aggregated value $\nu = \prod_i \Gamma_i = \prod_i g^{x_i y_i} g^{v_i}$ can be calculated by every participant or observer.
7. Since $\prod_i g^{x_i y_i} = 1$, by product associativity $\nu = \prod_i g^{v_i} = g^{\sum_i v_i}$ and therefore the aggregated value can be retrieved by solving the discrete logarithm equation by exhaustive search.

To ensure the correct operation of the above scheme, the broadcast value at point 3 should be checkable for actual knowledge of the exponent $x_i$ by agent $i$. Furthermore, the broadcast value at point 5 should be checkable for well-forming, i.e., for the validity of the selected value $v_i$. These tasks, in the original paper, are accomplished by accompanying the broadcast values with appropriate zero-knowledge proofs, but in this paper we assume

that the participants will correctly form their values. It is further assumed that each participant registered to a round will always broadcast both values in a timely manner—which is needed for the scheme to function. It is worth discussing point 7 in a bit more detail. Finding an arbitrary discrete logarithm is a notoriously intractable problem to solve in groups such as $G$, and this property is indeed what makes cryptography schemes such as this one secure. Nevertheless, the task of brute-forcing the aggregate value is possible provided that the values of the votes $v_i$ are not exceedingly big. In our case, we can use discretized energy values of resolution 0.01 kWh. In this way, the aggregated value of each LEM participant (a household or a public building with no industrial-scale, power-intensive request) will be of the order $\approx 1 \times 10^4$. In this way, with communities realistically comprising $\approx 1 \times 10^2$ participants, the aggregate values to check will be at most of $\approx 1 \times 10^6$. This process can be made more efficient in the typical case by adopting the heuristic of checking the most probable values first by keeping appropriate statistics on the previous rounds. Furthermore, the aggregated, encrypted value is to be considered a "receipt"; it is not essential to perform the modular logarithm every single time, but by keeping the encrypted value, anybody retains the faculty to check the correctness of market execution, should he find reason to do so. In other words, the "decoding" effort associated with step 7 in the procedure above can be largely skipped in normal situations and can be done afterwards, if needed.

In order to be able to execute the billing correctly, the community administrator must also know the private values of the individual participants. To do so, they can be encrypted with any symmetric cryptography scheme, so that their value is obfuscated on-chain, with the administrator holding all the keys. The administrator, therefore, can calculate the aggregate value directly, without having to perform the decoding. Nevertheless, the desired effect of making the computation trustless is mantained, as anybody can check through the homomorphic scheme the correspondence between the value that the admin uses for calculating the price in a certain time slot and the aggregated value. In case of mismatch, the administrator can simply use the aggregated, trustless value, but this problem is of secondary importance, as the "sincerity" of the agents in declaring their consumption/production value must be enforced at the hardware level, because the data are used for billing.

*6.2. Overhead Analysis*

Homomorphic encryption schemes carry a significant computational effort [28]. In this section, the privacy-preserving scheme influencing the computational burden is analyzed. We run a simulation of the execution of the protocol written in pure Python language for various initial parameters. We vary the length of $g$ in $512, 768, 1024$ bits, and the number of market participants from 10 to 80, in steps of 10. For each of these configurations, we run 100 attempts, and the $v_i$ is uniformly randomly chosen from $1, \ldots, 100$. The code for this analysis was run on Python 3.7 on the *ARM Cortex-A53 @1.20GHz*, the CPU of the Strato Computing Module installed in LIC pilot.

In Figure 6, the graph on the left shows the average computational time employed per participant to execute the full process of vote casting, included the computation of the encrypted aggregate (but not the decoding), and the on the right reports the average time for the discrete logarithm decoding as the maximum $\sum_i v_i$ increases. For the decoding part, in order to provide a comparison with more powerful hardware that could be used for the separate, a posteriori verification of the aggregate as explained above, the tests were also run on a commercial laptop CPU, the *Intel Core i7-7600U @2.80GHz*. In the right part of Figure 6, the results of this analysis are shown.

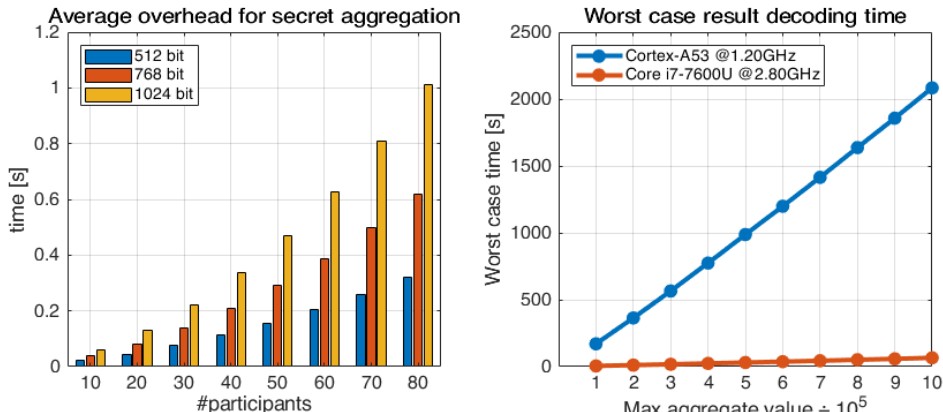

**Figure 6.** Testing of overhead and decoding times. The overhead increases for bigger (more secure) groups *G* and superlinearly with the number of participants, but remains manageable at realistic sizes. The decoding part is heavy on the Strato hardware, but it is not necessary to perform it at every market execution, and it can be executed on other hardware.

The computational time associated with the anonymous aggregation is not negligible. With a community size of 80 participants, and choosing a 1024-bit group generator *g*, each computing module spends about 1 s of additional time due to the execution of the cryptographic protocol. Nevertheless, because the aggregation happens in each 15 min time slot, this task remains more than manageable with a realistic community size. Decoding the final result, i.e., computing the discrete logarithm, is rather intensive for the reasons explained above. With a maximum admissible $\sum_i v_i$ of $10^6$, in the worst case (which means that the correct value is the last to be tried) this task takes about 2080 s on the *Cortex-A53*, with the time increasing linearly for even bigger values. As the aggregated encrypted values can be considered as a "receipt", as explained above, to perform a verification only if needed, this is not considered a problem. The same task performed on the *Intel Core i7-7600U @2.80GHz* CPU takes about 5 s. Therefore, from the experiments performed, the computational burden was considered acceptable. It is also worth noting that the cryptography would have to be broken every time in order to obtain a single datapoint for a user, and therefore even a 512-bit group could be considered secure enough for this application. For a case such as LIC, with $\approx 20$ participants, the computational times are, in any case, well below 0.2 s.

### 6.3. Deployment

The production version of the anonymization protocol was implemented in the Go language and deployed on the Strato modules in LIC, demonstrating the functionality of the scheme proposed.

### 6.4. Applicability of the Protocol

The described protocol performs a very well-defined primitive, which is the private and auditable aggregation of numeric values. It is therefore applicable to computations of prices, financial incentives, or other relevant quantities that depend only on the community-wide aggregation of consumption and production data. Thus, it appears that it could be widely useful provided that the participants are not treated in a distinct and individual way (which, as remarked in [2], could happen for theoretical congestion-dependent tariffs, but this type of mechanism would go against the fairness principles typically embraced by regulators). As an example, in the Italian case, communities may receive an incentive from the *Gestore dei Servizi Elettrici* publicly owned society, calculated on an hourly basis depending on self-consumption, that is, $min(E_p, E_c)$ [29]. The computation of this value could be accomplished in a verifiable way with the method illustrated here.

### 7. Conclusions and Future Work

In this paper, we presented the Lugaggia Innovation Community with its LEM implementation based on automated market making. The transactions and communication mechanisms for the LEM are executed on a dedicated Cosmos sidechain. Because the data about production and consumption of the participants are central to the economic exchange, the problem of securing this data arises, with the natural pseudonimity coming from the blockchain framework still presenting potential exploitability. For computing the applied market prices for each 15-minute period, only an aggregate quantity is needed, namely, the algebraic sum of production/consumption of the individual participants. Therefore, in this paper we proposed to use the secure, anonymous tallying protocol presented in [26], based on homomorphic encryption, which allows one to obtain the desired quantity without revealing the individual data points. We also suggested to skip the final, expensive step of the protocol, as it is needed to decode the sum only in case of conflicts or if a correctness check is desired by some party involved; the aggregated encrypted quantities are kept as "receipts". Albeit trusting the administrator of the community with the correctness of the computation is no longer needed, for billing reasons, the administrator of the community still needs to know the individual values. In the solution presented here, the individual quantities are sent to the administrator with symmetric encryption, so the administrator is trusted with the secure storage of the keys and the knowledge of the private data. A possible solution to remove this need would be to implement a mechanism based on zero-knowledge proofs to reveal to the administrator, for example, only the total monthly consumption. An interesting further development, therefore, would be to investigate such mechanisms in the context of ECs.

**Author Contributions:** Conceptualization, D.S. and V.M.; software, D.S. and F.R.; investigation, F.R., D.S. and L.N.; project administration, V.M.; visualization, F.R. and L.N.; writing, F.R., L.N., D.S. and V.M. All authors have read and agreed to the published version of the manuscript.

**Funding:** This research was funded by Horizon 2020 Framework Programme grant number 864319, Bundesamt für Energie grant number SI/501840.

**Institutional Review Board Statement:** Not applicable.

**Informed Consent Statement:** Not applicable.

**Data Availability Statement:** Not applicable.

**Acknowledgments:** This work has been supported by PARITY project funded by the European Union's Horizon 2020 Framework Programme for Research and Innovation under grant agreement no. 864319. The authors would like to thank all partners that have contributed to the work package on Local Flexibility Business and Market Models. This work has been also supported by the Swiss Federal Office of Energy, with the Lugaggia Innovation Community project, project number SI/501840.

**Conflicts of Interest:** The authors declare that they have no known competing financial interests or personal relationships that could have appeared to influence the work reported in this paper.

### Abbreviations

The following abbreviations are used in this manuscript:

| | |
|---|---|
| AEM | Azienda Elettrica di Massagno |
| AMM | Automated Market Making |
| BFT | Byzantine-Fault-Tolerant |
| CEC | Citizen Energy Community |
| CEP | Clean Energy Package |
| CPU | Central Processing Unit |
| DDH | Decisional Diffie–Hellman (assumption) |
| DSO | Distribution System Operator |
| EC | Energy Community |
| EU | European Union |

| FADP | Federal Act on Data Protection |
|------|-------------------------------|
| GDPR | General Data Protection Regulation |
| IoT | Internet of Things |
| LEM | Local Energy Market |
| LIC | Lugaggia Innovation Community |
| POD | Point of Delivery |
| PoS | Proof of Stake |
| PV | Photovoltaic |
| REC | Renewable Energy Community |
| RES | Renewable Energy Sources |
| SCC | Self Consumption Community |
| V2G | Vehicle-to-Grid |
| VPN | Virtual Private Network |
| ZEV/RCP | Zusammenschluss zum Eigenverbrauch |

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
