# Peer review of "Privacy and Auditability in the Local Energy Market of an Energy Community with Homomorphic Encryption"

_energies, doi:10.3390/en15155386_

Round 1
Reviewer 1 Report
The authors have presented an interesting topic. The manuscript could be accepted, but it needs some revision.
In the abstract, the abbreviation should be avoided.
Also, the abbreviation should be defined when the first time is mentioned. For example, on page 1, GDPR and FADP should be defined.
Page-1 line: 21-22 – the last part of the sentence: ”The Lugaggia Innovation Community (LIC) was born as a Self Consumption Community (SCC), with the sponsorship of the local DSO, Azienda Elettrica di Massagno (AEM). For this Energy Community, a Local Energy Market (LEM) was established, backed by a blockchain framework based on Cosmos [1], as extensively introduced in a first paper [2] authored (among others) by three of the authors of the present paper.” should be deleted.
Pg:3 l:129: The community comprises 18 residential houses, 4 of which four are equipped with roof photovoltaic panels for a total of 37 kWp, and a kindergarten, in which a battery with a capacity of 60 kWh and a 27 kWp photovoltaic system were installed.
The whole manuscript should be written in thrid-person.
Author Response
Dear Reviewer #1, thank you for your valuable review.
The abbreviations in the abstract were removed, and all acronyms are now defined at first use (and mirrored in the final abbreviation table). We also integrated your edits.
As far as the usage of the third person is concerned, we agree that in some cases the use of the first person is an avoidable aesthetic choice (e.g. in this section, we review...). We changed those instances. Nevertheless, we feel that in other, more substantial cases the usage of the first person better conveys intentionality and active choice (e.g.: we suggest...), and we would like to keep those sentences as they are. Furthermore, the only way to change such sentences is to use a passive form, which is discouraged in many academic style guides when active action is involved (for example Elsevier here https://scientific-publishing.webshop.elsevier.com/manuscript-review/using-active-and-passive-voices-academic-writing/ ).
Reviewer 2 Report
Interesting paper, with interesting application of the sidechain. Optical probes for the collection of meter data can be inaccurate and you should clarify whether your probe was a simple pulse detection probe or a more complex IEC 62056 compliant optical meter reader, which requires summation within the Strato Pi. Application to other communities not operating as single point of connection (ZEV) is also of interest.
Line 71: missing reference
Author Response
Dear Reviewer #2, thank you for your valuable review.
The probe employed is an IEC 62056 compliant optical reader, so we added more information in the hardware description.
We also added the missing reference to the Swiss energy law on line 71, which was an oversight.
We agree that clarifying the applicability of the method on different kinds of phisically connected communities is interesting, and therefore we added a paragraph (6.4) on this topic.